# Hashimoto’s Thyroiditis Does Not Influence the Malignancy Risk in Nodules of Category III in the Bethesda System

**DOI:** 10.3390/cancers14081971

**Published:** 2022-04-13

**Authors:** Dorota Słowińska-Klencka, Bożena Popowicz, Mariusz Klencki

**Affiliations:** Department of Morphometry of Endocrine Glands, Medical University of Lodz, Pomorska Street 251, 92-213 Łódź, Poland; b.popowicz@umed.lodz.pl

**Keywords:** thyroid, cancer, FNA, Bethesda system, category III, AUS, FLUS

## Abstract

**Simple Summary:**

Thyroid nodules in patients with Hashimoto’s thyroiditis (HT) have a high prevalence of equivocal cytology, especially category III of the Bethesda system. The aim of this study was to evaluate the risk of malignancy (RoM) in category III thyroid nodules in patients with and without HT, and to analyze whether obtaining category III with a repeat FNA increases RoM. We found that RoM in category III nodules was not affected by the presence of HT. In patients with nodules that show nuclear atypia, irrespective of the presence of HT, there are stronger indications for surgical treatment than in patients with nodules presenting architectural atypia. The chances of obtaining category III once again, with repeat FNA, are higher in patients with HT than without it. If the repeated diagnosis of category III results again from the architectural atypia, without signs of nuclear atypia, then the risk of malignancy does not increase.

**Abstract:**

The aim of this study was to evaluate the risk of malignancy (RoM) in category III thyroid nodules of the Bethesda system in patients with and without Hashimoto thyroiditis (HT) and to analyze whether obtaining category III with a repeat FNA (rFNA) increases RoM. The study included 563 HT and 1250 non-HT patients; rFNA was performed in 349 and 575 patients, and surgical treatment in 160 and 390, respectively. There was no difference in RoM between HT and non-HT patients in the whole examined population (lower limit of RoM), nor in operated patients (upper limit of RoM), HT: 5.0–17.5%, non-HT: 4.7–15.1%. RoM in patients with AUS nodules (with nuclear atypia) was similar in both groups (HT: 21.7–40.0%, non-HT: 16.9–41.4%), as it was in patients with FLUS nodules (with architectural atypia) (HT: 3.5–13.3%, non-HT: 4.0–13.0%). In patients from both groups together, with category III diagnosed twice and AUS identified at least once, RoM was 16.7–50.0% and it was higher than that in patients with FLUS nodule diagnosed twice: 3.2–13.0% (*p* < 0.005). Concluding, RoM in category III nodules is not affected by the presence of HT. Subcategorization of category III nodules (FLUS vs. AUS) may provide guidance toward further follow-up or surgery in both groups.

## 1. Introduction

Preoperative differentiation between benign and malignant thyroid nodules is still a challenging task. The main examination used for this purpose, fine-needle aspiration biopsy (FNA), does not provide an unequivocal answer in many cases. There are only two unambiguous categories (category II: benign lesions; category VI: malignant neoplasm) among the six categories of thyroid FNA outcomes identified in the Bethesda System for Reporting Thyroid Cytology (BSRTC) [1,2]. Other categories refer to smears that lack a satisfactory number of cells (category I–non-diagnostic), or to smears presenting an ambiguous microscopic image (category III: follicular lesion of undetermined significance-FLUS or atypia of undetermined significance-AUS; category IV: suspicious for a follicular neoplasm; category V: suspicious for malignancy). Category III is the most problematic among these three equivocal categories, as it is characterized by the highest diversity in terms of the estimated risk of malignancy (RoM), which ranges between 2% and 81% [2,3,4,5,6,7,8,9]. Initially, category III was meant to include less than 7% of FNA results, and its RoM was to be lower than 15% [1]. Accordingly, the standard management was to consist of the repeat FNA with molecular testing when available. Over time, those assumptions became questionable. In some centers, the frequency of category III was significantly higher and its RoM varied according to the nature of the atypia, reaching over 40% or even 80%, particularly in the case of nodules with nuclear atypia, commonly referred to as AUS [2,3,4,5,6,7,8,9]. Consequently, in centers where its RoM is high, category III becomes an obvious indicator for surgical treatment. On the other hand, in centers where its RoM is low (<10%) [3,10,11,12,13], the question arises: should conservative treatment be abandoned and surgical treatment be performed when a repeat FNA brings a category III result again (in as much as 1/3 of cases) [3,12]? That problem is of special significance in patients with Hashimoto’s thyroiditis (HT). They were shown to have a higher prevalence of indeterminate cytology in thyroid nodules when compared to patients without HT [14,15,16]. That refers particularly to category III. Our previous analysis indicated that patients with HT cytological outcomes of category III constituted as much as 15.6% of all FNA and 17.9% of diagnostic FNA [17], over twice as much as in the general population examined at our center—with or without HT (6.4% of diagnostic FNA) [18]. The suspicious cytology may be more common in HT patients because of postulated higher incidence of papillary carcinoma (PTC) in that cohort [15,19]. Another reason might be a difficulty in the interpretation of microscopic images with coexisting HT due to the common anisocytosis of thyroid follicular cells, their oxyphilic metaplasia, and variable inflammatory infiltration [14,15,16,20]. The above-mentioned difficulties in cytological diagnosis inclined us to evaluate the RoM in nodules of category III in BSRTC, separately for AUS and FLUS nodules, in relation to the status of HT occurrence in a patient. Another aim was to analyze whether obtaining category III with the repeat FNA increases the risk of malignancy in HT or non-HT patients.

## 2. Materials and Methods

### 2.1. Patients

FNA examinations were performed in a single center, in the years 2010–2021 (up to June), in patients referred from outpatient endocrinology clinics. The study included all patients who had a diagnosis of category III of BSRTC in that period and a known status of HT presence. The clinical diagnosis of HT was based on clinical symptoms, levels of anti-thyroid antibodies, serum concentrations of thyroid-stimulating hormone and thyroid hormones, ultrasound (US) examination, as well as microscopic examination. A detailed description of ultrasonographic features which we considered characteristic of HT was shown in our previous study [17]. We consider the following cytological image as typical of HT: inflammatory cells (lymphocytes, plasmacytes predominating in the smears) and thyroid follicular cells (usually scattered or clustered in small groups) showing anisocytosis and frequently oxyphilic metaplasia. Because generally fewer than 30% of patients with HT present all the above-mentioned HT features [21,22,23], we adopted minimal criteria for the diagnosis of HT. All patients included in HT group had a clinical diagnosis of HT confirmed with: elevated serum levels of anti-thyroid peroxidase antibodies or characteristic features of HT in the US or microscopic (cytological or histopathological) examination. On the other hand, patients included in the non-HT group did not show any hormonal, serological or morphological features of HT. Exclusion criteria were: (a) surgical or radioiodine thyroid treatment in the past, (b) positive irradiation history in the neck region, (c) positive history of Graves’ disease or increased concentration of antibodies against the TSH receptor.

FNAs were performed following regular procedures on thyroid nodules with a diameter of at least 5 mm (and usually over 1 cm) which showed at least one malignancy risk factor (US or clinical), according to the recommendations in effect in our country [24,25]. Biopsies were US-guided with the use of Aloka Prosound Alpha 7 ultrasound system, ALOKA co. Ltd., Tokyo, Japan and a 7.5–14 MHz linear transducer. At least two aspirations of a nodule were conducted in all cases. Smears were stained with hematoxylin and eosin after the fixation with 95% ethanol. Cytological diagnoses were formulated according to the BSRTC classification in the version prior to the modification in 2017 [1,2]. In this version, smears presenting nuclear features of PTC were excluded from the category IV: suspicious for a follicular neoplasm. FLUS was diagnosed when features from the borders of categories II and IV were identified. The diagnosis of AUS was made when local features suggestive of PTC (nuclear grooves, enlarged nuclei with pale chromatin and alterations in nuclear contour and shape) were present in an aspirate that was otherwise benign in microscopic appearance. AUS was also diagnosed in specimens with limited cellularity but with nuclear atypia. A detailed description on the classification of nodules into specific diagnostic categories of the Bethesda system, as well as the risk of malignancy related to particular categories at our center were presented in our earlier reports [18,26]. All examined patients gave their informed consent to perform FNA of the thyroid.

The risk of malignancy in nodules of category III was assessed with the use of outcomes of repeat FNA (rFNA) and postoperative histopathologic examinations. The surgical treatment was performed based on the patient’s preference or due to the large size of the goiter or the presence of other clinical risk features, including the suspicious ultrasound image of a nodule or as a consequence of the alarming outcome of rFNA. The histopathologic examination was performed according to the standard procedures. Diagnoses were formulated according to the WHO classification of thyroid tumors that was in effect at the time. We did not review histopathological specimens to reveal cases of NIFTP.

Overall, the study included 1813 patients, i.e., 563 patients with HT (HT group) and 1250 patients without HT (non-HT group) (Table 1).

The mean age of patients in HT group was lower than in non-HT group, as was the percentage of males. The fraction of patients with category III diagnosis in more than one nodule was similar in both groups and did not exceed 5% (HT group: 24/4.3% patients with two category III nodules and 3/0.5% patients with three such nodules; non-HT group: 51/4.1% and 4/0.3%, respectively). In three patients the initial FNA diagnosed AUS in two coexisting nodules (two patients in HT group and one non-HT patient), and in one patient of HT group AUS was identified in three nodules. In four other patients an AUS nodule was accompanied by a FLUS nodule (three patients in HT group and one non-HT patient)–those patients were regarded as patients with an AUS nodule for the purpose of the study. In other patients with multiple nodules of category III only FLUS nodules were observed. The percentages of patients with AUS nodules and of AUS nodules themselves were similar in both groups, with an insignificant lead of HT group. Nodules’ size was usually ranged from 11 to 20 mm. Sizes of nodules in HT group fell into that range more often than nodules in non-HT group, while nodules with a diameter >30 mm were more frequent in non-HT group.

### 2.2. Analyzed Variables

At first the outcomes of rFNA were analyzed in both groups of patients. The distribution of rFNA results among particular categories of BSRTC was evaluated. Such an analysis was also performed separately for AUS and FLUS nodules in both groups. The outcomes of rFNA were analyzed on a per-patient basis as well as a per-nodule basis (see tables in the Appendix A). Only rFNAs performed within two years since the first FNA were considered. When several rFNAs were done in that period, the outcome with the highest category of BSRTC was analyzed. Analogically, for the sake of per-patient analysis, in patients with multiple nodules examined, the nodule with the highest category of BSRTC was considered. The distribution of rFNA outcomes among particular BSRTC categories was also analyzed separately in the group of operated patients.

Next, the results of postoperative histopathological examinations were evaluated. The RoM was estimated in patients with category III nodules of both groups by establishing its lowest and highest possible values. The lower limit of RoM was defined as a quotient of the number of histopathologically verified cancers and the total number of all patients with category III nodules (operated and not operated). The upper limit of RoM was defined as a quotient of the number of histopathologically verified cancers and the number of operated patients. In both cases cancers revealed in nodules other than category III nodules were not considered. Analogically lower and upper limits of RoM were established for patients from both groups operated after the initial FNA and those operated after rFNA which again indicated category III. All those analyses were also performed for patients with AUS or FLUS nodules separately in both HT and non-HT groups. Patients were considered as having an AUS nodule if the AUS subcategory was diagnosed in any FNA (initial or repeat one). The types of identified cancers were also compared in both groups.

The study protocol was approved by the local Bioethics Committee. According to the Committee’s approval neither patient’s approval nor the informed consent for our review of patients’ clinical data and FNA results were needed.

### 2.3. Statistical Evaluation

The statistical significance of obtained results was evaluated with the use a dedicated data analysis software system, Dell Statistica, version 13, Dell Inc. (2016), Round Rock, TX, USA. Frequency distributions were compared with chi2 test (modifications appropriate for the number of analyzed cases were applied). The comparison of continuous variables between groups was made with the use of Kruskal–Wallis test. The value of 0.05 was assumed as the level of significance.

## 3. Results

### 3.1. Outcomes of Repeat FNA

The repeat FNA was performed in 349 patients in the HT group and 575 patients in the non-HT group, more often in the HT than non-HT group (62.0% vs. 46.0%, *p* < 0.0001). Those repeat examinations allowed reevaluation of 367 and 597 nodules classified into category III by the initial FNA in both groups, respectively (Table 2).

In both groups the most common outcome of rFNA was that of category II. Table 3 and Table 4 show the distribution of categories of rFNA outcomes in patients from both groups, while Appendix A show analogous data in a per-nodule manner. Regardless of the mode of analysis, the same regularities were observed.

The frequency of category II in rFNA outcomes was significantly lower in patients in the HT group than in the non-HT group (48.7% vs. 55.7%, *p* < 0.05), while the frequency of category III was higher (42.1% vs. 28.0%, *p* < 0.0001). The main source of those differences was the specific distribution of rFNA outcomes in patients with AUS nodules in the HT group. Category II was formulated in those patients two-times less often (26.9%) than in other patients in both groups (non-HT group, patients with AUS nodules: 59.4%, *p* = 0.0135; patients with FLUS nodules: 55.4%, *p* = 0.0044; HT group, patients with FLUS nodules: 50.5% *p* = 0.0209). Accordingly, category III in rFNA was observed more often in patients in the HT group with AUS nodules than in other patients, both those in the non-HT group with AUS nodules (61.5% vs. 25.0%, *p* = 0.0050) and those with FLUS nodules, regardless of the group (HT: 40.5%, *p* = 0.0371, non-HT: 28.2%, *p* = 0.0003). Category III in rFNA was also more frequent in patients with FLUS nodules in the HT group than in the non-HT group (40.5% vs. 28.2%, *p* = 0.0002).

In both groups, in patients with FLUS nodules, the diagnosis of AUS in rFNA was rarely observed—HT: 3.1% (4 out of 131); non-HT: 1.3% (2 out of 153). The repetition of AUS diagnosis in rFNA was observed in the HT group more often than in the non-HT group (46.1% vs. 12.5%, *p* = 0.0106). There was no category IV result in rFNA, in any patient in the HT group, while in patients in the non-HT group, such a diagnosis was observed in the case of FLUS nodules only. Diagnoses of category V or VI were observed in only 2.3% of rFNA outcomes in the HT group and 1.5% of outcomes in the non-HT group (NS). Non-diagnostic smears in rFNA were more common in the non-HT group than in the HT group (13.4% vs. 6.9%, *p* = 0.0021).

### 3.2. Surgical Treatment without or after Repeat FNA

The surgical treatment was performed in 160 (28.4%) patients in the HT group and 390 (31.2%, NS) patients in the non-HT group. Among operated patients, the frequency of patients subjected to rFNA in the HT group was higher than in the non-HT group: 35.6% (57 out 160) vs. 24.1% (94 out of 390); *p* = 0.0060 (Table 2). Appendix A shows the distribution of surgically treated patients and their nodules between the examined groups, regarding their rFNA status and the initial diagnosis (FLUS vs. AUS).

Among patients who had rFNA performed, the rates of surgical treatment were the same and equaled 16.3% (57 out of 349 in the HT group and 94 out of 575 in the non-HT group). Analysis of the distribution of rFNA outcome categories in operated patients showed that, in both groups, the most common category was category III, but in the HT group, its frequency was two-times higher than in the non-HT group: 70.2% vs. 41.5% (*p* = 0.0066). Category II was the second most common and it was more frequent in the non-HT group than the HT group: 29.8% vs. 12.3% (*p* = 0.0135), respectively. The frequency of other BSRTC categories of rFNA outcomes in operated patients did not differ significantly between the HT and non-HT groups (categories I: 3.5% vs. 11.7%; IV: 0.0% vs. 8.5%; V: 10.5% vs. 6.4%; VI: 3.5% vs. 2.1%, respectively).

Cancers were diagnosed in 87 (15.8%) of the operated patients, 28 (17.5%) in the HT group and 59 (15.1%, *p* = 0.4887) in the non-HT group. There was no significant difference in the incidence of PTC between both groups either (Appendix A). The incidence of PTC in the non-HT group was nearly significantly higher in patients with AUS nodules than FLUS nodules (83.3% vs. 48.9%, *p* = 0.0693), while in the HT group, it was similar (70.0% vs. 66.7%, *p* = 0.8093).

We did not find any significant difference in the mean age of patients with cancers and those with benign lesions in either group, HT: 49.8 ± 15.9 vs. 53.3 ± 13.1 (*p* = 0.2189); non-HT: 55.6 ± 14.7 vs. 55.1 ± 12.5 (*p* = 0.7896), respectively. Similarly, there was no significant difference in the rate of males among patients with cancers and benign lesions in either group, the percentages of males were 7.1% vs. 3.8% (*p* = 0.7797) in the HT group, and 20.3% vs. 16.0% (*p* = 0.4113) in the non-HT group, respectively.

### 3.3. Risk of Malignancy

We did not find any significant difference in the malignancy rate between patients of the HT and non-HT groups in the whole examined population (operated and not operated), the lower limit of RoM, nor in operated patients, the upper limit of RoM, and the range of RoM was 5.0–17.5% in the HT group, and 4.7–15.1% in the non-HT group. Table 5 shows data on RoM in patients, and Appendix A shows analogous data on RoM in the examined nodules.

No significant difference between the groups in the lower or upper limit of RoM was found, either in patients operated directly after the first FNA of category III (HT: 7.9–16.5%, non-HT: 5.8–13.2%), or in patients operated after rFNA, in which category III had been diagnosed again (HT: 4.1–15.0%, non-HT: 5.0–20.5%). Nodules with a diameter over 30 mm in patients operated on directly after the first FNA were more common in the non-HT group than the HT group (29.1% vs. 12.6%, *p* = 0.0007), while nodules with a diameter under 20 mm were less frequent (50.4% vs. 67.9%, *p* = 0.0048). We did not find any significant difference between the HT and non-HT groups in the RoM, despite the preponderance of AUS nodules in patients in the HT group (HT: 15.6% vs. non-HT: 7.4%, *p* = 0.0034 in operated patients and HT: 8.2% vs. non-HT: 5.7%, *p* = 0.0458 in all studied patients), which were characterized by a higher RoM than FLUS nodules. The RoM in patients with AUS nodules was similar in the HT and non-HT groups (HT: 21.7–40.0%, non-HT: 16.9–41.4%, *p* = 0.5130 for the lower limit of RoM and *p* = 0.9181 for the upper limit of RoM), as it was in patients with FLUS nodules (HT: 3.5–13.3%, non-HT: 4.0–13.0%, *p* = 0.5633 for the lower limit of RoM and *p* = 0.9514 for the upper limit of RoM) (Table 5).

### 3.4. Influence of Persistent Category III Outcome in rFNA on the Risk of Malignancy

When the diagnosis of category III was formulated again in rFNA, it did not significantly affect the upper limit of RoM, in comparison with that observed in patients operated on without rFNA, in either group. In the HT group, that limit was 1.5 percentage points lower (15.0% vs. 16.5%, *p* = 0.8260), and in the non-HT group, 7.3 percentage points higher (20.5% vs. 13.2%, *p* = 0.2149) than in patients in the same group, operated on without rFNA. The lower limit of RoM in patients operated on after one or two diagnoses of category III did not differ significantly either, HT: 7.9% vs. 4.1%, *p* = 0.1399; non-HT: 5.8% vs. 5.0%, *p* = 0.6889. Significant differences were revealed only when the subtypes of category III identified in the initial and repeat FNA were analyzed in the whole examined population. In patients with category III diagnosed twice and with AUS identified at least once (in the initial or repeat FNA), the malignancy rate (the upper limit RoM) was 50.0% (Table 5). That rate was not significantly different from the one found in patients with AUS nodules operated on without rFNA: 34.3% (*p* = 0.3660), but it was significantly higher than the one in patients with a FLUS nodule diagnosed in both FNAs: 13.0% (*p* = 0.0042). A significant difference in the lower limit of RoM was also observed between patients with category III diagnosed twice and AUS identified at least once and patients with FLUS nodules diagnosed in both FNAs (16.7% vs. 3.2%, *p* = 0.0008). The diagnosis of FLUS in rFNA did not change the malignancy rate from that observed in patients operated on without rFNA, neither in the case of a FLUS nodule in the initial FNA (13.0% vs. 12.1%, *p* = 0.8243), nor in the case of an AUS nodule (33.3% vs. 34.3%, *p* = 0.9734). Accordingly, the malignancy rate in patients with an AUS nodule operated on without rFNA was higher than the one in patients with a FLUS nodule diagnosed in both FNAs (34.3% vs. 13.0%, *p* = 0.0108). Interestingly, in patients with an AUS nodule in the initial FNA, confirmation of that diagnosis in rFNA was related to the 100.0% malignancy rate, but this was the case in only two patients (one in each group).

When patients (or nodules) in the HT and non-HT groups were analyzed separately, similar regularities were observed (Table 5 and Appendix A), but they were generally not statistically significant. Notably, the malignancy rate of FLUS nodules in the HT group was several percentage points lower when FLUS was found again in rFNA, in comparison with patients with FLUS nodules operated on without rFNA (9.1% vs. 13.5%, *p* = 0.7294).

## 4. Discussion

Hashimoto thyroiditis, also called chronic lymphocytic or autoimmune thyroiditis, is the most common autoimmune endocrine disease, as well as the most common cause of hypothyroidism. This inflammation results in significant changes to the morphology of the thyroid gland that affect both its ultrasound and cytological images. Ultrasound examination may sometimes be puzzling due to the high heterogeneity and hypoechogenicity of the thyroid gland. Evaluation of cytological specimens is difficult because there are reactive changes in the course of HT that may obscure subtle features of PTC or—contrarily—may lead to the over-diagnosing of cancers [16,17,20]. Consequently, FNA in patients with HT brings equivocal results, mainly of category III in the BSRTC system, more often [14,15,16]. Our present study shows that nodules in that category, in patients with HT, are also characterized by a higher frequency of category III results of rFNA (42% in HT group vs. 28% in non-HT group). Previous studies showed a wide range in the rate of Bethesda category III being categorized as persistent on repeat FNA, from about 10% to around 50% [27,28,29,30,31,32,33]. However, none of those studies analyzed the possible effect of coexisting HT.

Our data showed that the higher frequency of category III in nodules of patients with HT was not related to a higher rate of malignancy. The RoM in category III nodules was in the range of 5.0% to 17.5% in the HT group and 4.8% to 15.1% in the non-HT group. Those ranges were determined by the risk estimated for all patients, including those not operated on (the lower limit) and by the frequency of malignancy in patients who were operated on (the upper limit). In both cases, patients in whom rFNA had changed the BSRTC category were also included. That was a consequence of the specificity of category III, which is an indication for rFNA. In the case of other equivocal categories (IV and V), the RoM is usually related to the result of the last FNA performed before the surgery (it is frequently the only FNA). Our study did not show any difference in the RoM of AUS nodules in both groups. There was also no such difference in the case of FLUS nodules. However, we confirmed our earlier reports, as well as a number of other studies, showing that the RoM of AUS nodules (in the range of 18.8–40.7% in our study) is significantly higher than that of FLUS nodules (3.9–13.1% in our study) [4,5,7,27,33,34,35,36,37].

So far, only a few studies have addressed this subject. Most of them reported results concordant with our observations. Their authors indicated that HT was not a risk factor for thyroid malignancy nodules with category III. Cho et al. [38] found the same frequency of malignancy—48%—in patients with and without HT. Rotondi et al. [39] also reported similar values—11.0% vs. 13.4%, respectively. Suh et al. [40] and Mulder et al. [28] observed that the frequency of malignancy in patients with HT was over 10 percentage points lower than the one in patients without HT (Suh: 36.0% vs. 46.6%, Mulder: 44% vs. 60%). Ogmen et al. [41] found that in patients with nodules of category III, anti-thyroid antibody positivity rates were similar in patients with benign and malignant histopathology. Topaloglu et al. [42] did not report any direct association between HT and malignancy either, although they observed a higher frequency of malignancy in patients with HT (51.0% vs. 41.2% in non-HT patients) and a more frequent presence of thyroid autoantibodies in patients with malignancy.

However, there are reports with opposite conclusions. Silva de Morais et al. [14] showed that the relative risks of malignancy in nodules of category III were about 50% higher in patients with HT than in those without HT (the frequency of malignancy in those groups was 27.6% vs. 20.5%, respectively). Further, Huang et al. [43] found that anti-TPO concentrations were significantly correlated with the malignancy of category III nodules.

None of the mentioned studies analyzed the rates of malignancy in the subgroup of nodules with nuclear atypia, usually called AUS (alternatively, ‘AUS/FLUS—cannot exclude PTC’ or ‘AUS-N’ in some reports), or nodules with architectural atypia, usually called FLUS (but also ‘AUS/FLUS—cannot exclude follicular neoplasm’ or ‘AUS-A’) separately [4,7,29,35], despite the fact that marked differences in malignancy rates between AUS and FLUS nodules strongly suggest the need for subcategorization of thyroid nodules of category III. The relative frequency of nodules with nuclear atypia and those with architectural atypia within category III noticeably influences RoM of that whole category and it helps clarify observed differences in RoM between various centers. Most patients diagnosed at our center had been exposed to iodine deficiency for many years. In consequence, they are characterized by a high percentage of benign follicular lesions, presenting a cytological image often typical of FLUS. This subcategory predominates within category III in our material and determines the relatively low value of RoM of that category. Our earlier studies indicated that the relative frequency of AUS and FLUS nodules may also be affected by other factors, such as variable sensitivity of pathologists from endocrine and oncological centers to the degree of atypia [26]. Pathologists working at endocrine centers more often evaluate smears in which features of atypia may result from HT, hyperthyroidism, antithyroid agent use or radioiodine treatment. This relatively high influence of center-specific factors on the interpretation of rules for classification of nodules into category III comes from the definition of this category, which is not precise enough.

In our opinion, the subcategorization of category III nodules is also necessary for the reliable assessment of clinical significance of rFNA results, especially those classified again into category III. In fact, we did not find any difference in the frequency of malignancy in patients operated on after either single or double diagnosis of category III in FNA. Only a more detailed analysis of AUS and FLUS subcategories showed that in nodules classified into category III, both in first and repeat FNA, the distinction whether there were two FLUS diagnoses, or rather, at least one AUS diagnosis, actually mattered. The latter was associated with the lower limit of RoM, amounting to at least 15%—we found RoM ranges to be 15.0% to 42.9% in the HT group and 20.0% to 66.7% in the non-HT group. Such values justify surgical treatment without further FNA. The rate of malignancy in the case of nodules with FLUS diagnosed twice is lower than that of nodules with one or two AUS diagnoses and it is similar to the rates for nodules with a single FLUS diagnosis. Thus, the decision about surgical treatment in such patients (with repeated FLUS diagnosis) has to supported by other clinical factors. It is especially valid in patients with HT, as we even observed a decrease in RoM, by several percentage points, in such a case.

Most of the studies of other researchers, carried out without the subcategorization of category III nodules, did not show significant differences in malignancy rates of nodules in patients operated on with a single or double diagnosis of category III. Broome et al. [31] found nearly identical malignancy rates (about 15.4%). VanderLaan et al. [30] estimated the rates to be 41% and 43%, respectively; Ogmen et al. [41]—30.6% and 32.2%; Yoo et al. [32]—76.8% and 76.0%. In some other studies, the reported differences in the incidence of malignancy after one and after two category III diagnoses, as expressed in percentage points, were higher but still insignificant: Ho et al. [29] (38.6% vs. 26.3%), Sullivan et al. [44] (34% vs. 50%), Kuru et al. [45] (15.6% vs. 26%), Hong et al. [46] (61% vs. 50%), respectively.

There are also reports showing some significant differences in malignancy rates observed in patients operated on after a single category III diagnosis, in comparison to those operated on after two such diagnoses; however, they are somewhat contradictory. Some authors found lower malignancy rates in patients operated on directly after a single category III diagnosis: Kaliszewski et al. [47]—7.4% vs. 18.5%; Kaya et al. [48]—13.3% vs. 27.5%. Others reported increased rates of malignancy in that group of patients: Park et al. [49]—77.6% vs. 58.1%; Gweon et al. [50]—78.3% vs. 37.2%. It should be noted that, in the latter study, the nearly twofold difference in the malignancy rate was observed between patients operated on directly after the first biopsy and those operated on after rFNA, the results of which were classified not only into category III, but also IV, V and VI (the study included only one patient with a benign lesion diagnosis in rFNA). Both Park’s and Gweon’s studies were performed in Asia, where PTC is predominant over other thyroid cancers, and cases with nuclear atypia among category III smears are twice as common as in non-Asian populations. In consequence, the rate of malignancy is high in the Asian cohort [7]. PTC presents a characteristic ultrasound image, and this helped in the effective qualification of patients harboring the cancer for surgical treatment, just after a single FNA result of category III. In the group of patients referred to surgery after obtaining two diagnoses of category III, the percentage of follicular variant PTC among cancers was higher—and their ultrasound image was less suggestive [49]. In the population examined at our center, the percentage of PTC among cancers was markedly lower, especially in nodules with undetermined cytology [9]. The effectiveness of the assessment of ultrasound malignancy risk features in that group of nodules is also lower, but as we showed earlier, the diagnostic value of TIRADS systems is similar in patients with and without HT [51]. Therefore, our observation of a higher frequency of the surgical treatment performed directly after the first diagnosis of category III in patients without HT was probably a consequence of larger nodules’ sizes and not an especially suspicious ultrasound image.

We did not analyze the significance of results of repeat FNA, other than category III, because of the relatively small number of such cases in the examined groups, especially in the case of categories V and VI. Many studies have demonstrated the benefits of repeat FNA in reclassifying category III nodules into a category with a better-established malignancy rate and management [7,27]. Notably, just as in the case of category III, there are marked differences between diagnostic centers in observed malignancy rates in patients operated on after particular BSRTC categories of repeat FNA [7].

There are some limitations in our study. It was retrospective in design, and the clinical decision making (thyroid excision or repeat biopsy) was often influenced by patient preference, rather than the recommendations of endocrinologists. Additionally, the actual incidence of NIFTP and other borderline tumors could not be determined, as we did not review histopathological specimens. The advantages of our study include a high number of examined patients and a high number of patients operated on after repeat FNA classified into category III, in comparison to other published studies. Another distinctive feature is the separate analysis of patients with nodules presenting nuclear atypia and architectural atypia. The specification of RoM ranges, which approximate the actual risk of malignancy, is also advantageous. It was an attempt to balance two sources of bias. One is related to the fact that patients with nodules that show unfavorable clinical or sonographic features are more eagerly referred to surgical treatment. This bias increases the observed rate of malignancy. The second one comes from regarding all not-operated-on nodules as benign, which decreases the calculated rates of malignancy.

## 5. Conclusions

The risk of malignancy in category III nodules is not affected by the presence of HT. In both HT and non-HT patients, the subcategorization of category III nodules may improve patient follow-up procedures and provide guidance toward further follow-up or surgery. In the case of patients with nodules that show nuclear atypia, irrespective of the presence of HT, there are stronger indications for surgical treatment than in the case of patients with nodules presenting architectural atypia. In the latter group of patients, the decision about surgical treatment should be better supported by additional clinical factors, including the results of repeat FNA that show nuclear atypia or a higher risk BSRTC category. In patients with HT, the chances of obtaining category III once again with repeat FNA are higher than in patients without HT. However, if the repeated diagnosis of category III results from the architectural atypia without signs of nuclear atypia, then the risk of malignancy does not increase in either group. Consequently, it should not be a major reason for surgical treatment. Nodules with undetermined cytology constitute a specific diagnostic problem, especially in populations that have been exposed to iodine deficiency, where the fraction of PTC among thyroid cancers is lower. Such nodules rarely present typical sonographic risk features. In that specific group of nodules, additional guidance from molecular examination of material obtained with FNA would be especially valuable. Unfortunately, such examinations are limited by high costs and the lack of satisfactory validation in various populations. For now, we should remember not only about the increased incidence of PTC in patients with HT, but also about the possibility of overdiagnosis in the case of reactive changes present in smears from patients with HT.

## Figures and Tables

**Table 1 cancers-14-01971-t001:** Demographic data of the patients and distribution of nodules’ sizes in the HT and non-HT groups.

Variable	HT	Non-HT	*p*
Number of patients	563	1250	
Age, mean ± SD [years]	57.5 ± 14.3	59.7 ± 13.5	0.01
No./% of males	29/5.2	185/14.8	<0.0001
No./% of patients with more than one nodule of category III	27/4.8	55/4.4	0.7075
No./% of patients with AUS nodule in the first FNA	42/7.4	70/5.6	0.1280
Number of nodules	593	1309	
No./% of nodules ≤ 10 mm	95/16.0	162/12.4	0.0313
No./% of nodules 11–20 mm	345/58.2	627/47.9	<0.0001
No./% of nodules 21–30 mm	106/17.9	277/21.2	0.0979
No./% of nodules > 30 mm	47/7.9	243/18.6	<0.0001
No./% of AUS nodules in the first FNA	46/7.8	71/5.4	0.0498

**Table 2 cancers-14-01971-t002:** Data on frequencies of repeat FNA and surgical treatment in the HT and non-HT groups.

Variable	HT	Non-HT	*p*
No./% of patients who underwent repeat FNA	349/62.0	575/46.0	<0.0001
No./% of operated patients	160/28.4	390/31.2	0.2334
No./% of operated patients without or after repeat FNA	103/64.4 57/35.6	296/75.9 94/24.1	0.0060
No./% of operated patients with category III diagnosis in the repeat FNA	40/25.0	39/10.0	<0.0001
No./% of nodules examined with repeat FNA	367/61.9	597/45.6	<0.0001
No./% of excised nodules	172/29.0	411/31.4	0.2944
No./% of excised nodules without or after repeat FNA	111/64.5 61/35.5	309/75.2 102/24.8	0.0090
No./% of excised nodules with category III diagnosis in the repeat FNA	42/24.4	46/11.2	<0.0001

**Table 3 cancers-14-01971-t003:** Results of repeat FNA in patients from HT and non-HT groups.

Category of Repeat FNA	No./% of Patients with Repeat FNA		
	HT 349	Non-HT 575	*p*
I	24/6.9	77/13.4	0.0026
II	170/48.7	320/55.7	0.0404
III	147/42.1	161/28.0	<0.0001
IV	0/0.0	8/1.4	0.0269
V	6/1.7	7/1.2	0.5301
VI	2/0.6	2/0.3	0.9911

**Table 4 cancers-14-01971-t004:** Results of repeat FNA in patients of HT and non-HT groups for AUS and FLUS nodules separately.

Category of Repeat FNA	No./% of All Patients with Repeat FNA					
	HT			Non-HT		
	AUS 26	FLUS 323	*p*	AUS 32	FLUS 543	*p*
I	1/3.8	23/7.1	0.8166	4/12.5	73/13.4	0.9087
II	7/26.9	163/50.5	0.0209	19/59.4	301/55.4	0.4384
III	16/61.5	131/40.5	0.0371	8/25.0	153/28.2	0.6973
AUS	12/46.1	4/1.2	<0.0001	4/12.5	2/0.4	<0.0001
FLUS	4/15.4	127/39.3	0.0268	4/12.5	151/27.8	0.0907
IV	-	-		-	8/1.5	0.9322
V	1/3.8	5/1.5	0.9337	1/3.1	6/1.1	0.8546
VI	1/3.8	1/0.3	0.3432	-	2/0.4	0.1657

**Table 5 cancers-14-01971-t005:** Risk of malignancy (RoM) in patients from HT and non-HT groups with AUS and FLUS nodules analyzed in three sets of patients: (A) the whole examined cohort, (B) only patients without repeat FNA, and (C) only patients with category III in rFNA. RoM was evaluated as a range with its lower limit defined by the number of patients with cancer to the total number of patients (A1, B1 and C1) and its upper limit defined by the number of patients with cancer to the number of operated patients (A2, B2 and C2).

Set of Patients and RoM Range Limit	Risk of Malignancy (RoM) [%] (Number of Patients with Cancer)											
	All			HT			Non-HT			All		
	HT	Non-HT	*p*	AUS	FLUS	*p*	AUS	FLUS	*p*	AUS	FLUS	*p*
A1–lower limit all patients HT: 563, non-HT: 1250	5.0 (28)	4.7 (59)	0.8153	21.7 (10)	3.5 (18)	0.0013	16.9 (12)	4.0 (47)	<0.0001	18.8 (22)	3.8 (65)	<0.0001
A2–upper limit operated patients HT: 160, non-HT: 390	17.5 (28)	15.1 (59)	0.4887	40.0 (10)	13.3 (18)	0.0013	41.4 (12)	13.0 (47)	<0.0001	40.7 (22)	13.1 (65)	<0.0001
B1–lower limit all patients without rFNA HT: 214, non-HT: 675	7.9 (17)	5.8 (39)	0.2557	31.3 (5)	6.1 (12)	0.0003	18.4 (7)	5.0 (32)	0.0006	22.2 (12)	5.3 (44)	<0.0001
B2–upper limit patients operated without rFNA HT: 103, non-HT: 296	16.5 (17)	13.2 (39)	0.4021	35.7 (5)	13.5 (12)	0.0373	33.3 (7)	11.6 (32)	0.0046	34.3 (12)	12.1 (44)	0.0003
C1–lower limit all patients with category III in rFNA HT: 147, non-HT: 161	4.1 (6)	5.0 (8)	0.7088	15.0 (3)	2.4 (3)	0.0407	20.0 (2)	4.0 (6)	0.1317	16.7 (5)	3.2 (9)	0.0008
C2–upper limit patients operated after category III in rFNA HT: 40, non-HT: 39	15.0 (6)	20.5 (8)	0.5212	42.9 (3)	9.1 (3)	0.0911	66.7 (2)	16.7 (6)	0.1880	50.0 (5)	13.0 (9)	0.0042

Patients were considered as harboring an AUS nodule when AUS was diagnosed in at least one FNA (first or repeat one); rFNA—repeat FNA.

## Data Availability

The data presented in this study are available on request from the corresponding authors. The data are not publicly available due to patient privacy restrictions.

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
