# Peer review of "Hashimoto’s Thyroiditis Does Not Influence the Malignancy Risk in Nodules of Category III in the Bethesda System"

_cancers, 2022, doi:10.3390/cancers14081971_

Round 1
Reviewer 1 Report
This is a very interesting paper demonstrating that Hashimoto’s thyroiditis does not influence the malignancy risk in thyroid nodules of Bethesda III category. I have some minor comments.
- The great part of Material and Methods section, mainly "Patients" section including Table 1, presents the results of the study. Please place the results in the proper section, after creating adequate subsections as needed.
- No detail data on either US equipment used in the study or on FNAB procedures are provided. Please add proper information.
- Extensive language correction should be performed by a native speaker
Author Response
Thank you for positive comments on our study. We have followed all the detailed remarks:
- 1. The great part of Material and Methods section, mainly "Patients" section including Table 1, presents the results of the study. Please place the results in the proper section, after creating adequate subsections as needed.
According to the reviewer’s suggestion some text from the ”Patients” subsection of “Material and Methods” has been moved to the Results, including data on the frequency of repeat FNA (to the beginning of the “Outcomes of repeat FNA” subsection) and data on the frequency of surgical treatment in relation to repeat FNA (to the new subsection “Surgical treatment without or after repeat FNA”). Table 1 has been split into two separate tables with one containing data only pertinent to these presented in Material and methods, and the other showing data on the frequency of repeat FNA and surgical treatment. The latter has been located in the Results. Similarly, the reference to Figure S1 has been moved to that section too. We believe that the revised version is much more naturally structured – thank you for that remark.
- No detail data on either US equipment used in the study or on FNAB procedures are provided. Please add proper information.
The US examinations were performed with the use of the Aloka Prosound Alpha 7 ultrasound system, ALOKA co. Ltd., Tokyo, Japan with a 7.5–14 MHz linear transducer. The appropriate information has been added to the Material and methods section.
- Extensive language correction should be performed by a native speaker
The whole manuscript has been checked for errors as suggested.
Reviewer 2 Report
Słowińska-Klencka et al. aimed to evaluate the risk of malignancy (RoM) in category III thyroid nodules of Bethesda system in patients with and without Hashimoto thyroiditis (HT) and to analyze whether obtaining category III with a repeat FNA (rFNA) increases RoM. The study included 563 HT and 1250 non-HT patients. A repeat FNA was performed in 349 and 575 patients, while surgical treatment was done in 160 and 390 patients, respectively. As many other authors also Słowińska-Klencka et al. found out, that the frequency of malignancy was not different in patients with and without HT. Słowińska-Klencka et al. reported that chances for obtaining category III with repeat FNA are higher in patients with HT than without HT. Unfortunately, the results of the histological examination were not clearly presented. It is not clearly stated, how many patients had thyroid cancer. Only 87 as reported in table S4? Furthermore, the results of cytological analysis and the risk of malignancy are not presented clearly. But they are presented several times in different manners in the text and the tables. This makes the text very difficult to understand. My view is that the manuscript should not be published in the Cancers journal.
Author Response
Słowińska-Klencka et al. aimed to evaluate the risk of malignancy (RoM) in category III thyroid nodules of Bethesda system in patients with and without Hashimoto thyroiditis (HT) and to analyze whether obtaining category III with a repeat FNA (rFNA) increases RoM. The study included 563 HT and 1250 non-HT patients. A repeat FNA was performed in 349 and 575 patients, while surgical treatment was done in 160 and 390 patients, respectively. As many other authors also Słowińska-Klencka et al. found out, that the frequency of malignancy was not different in patients with and without HT.
The last sentence is unclear and not true. If the reviewer meant the rates of malignancy in patients with HT (in comparison with patients without HT) with no regard to other factors (like category of FNA outcome) we absolutely disagree. Actually, many investigators reported higher malignancy rates in patients with HT, mainly due to higher incidences of PTC. If the reviewer meant only patients with category III we also disagree. There are only a few reports focused on possible differences in malignancy rates in patients with category III in relation to their HT status. Their conclusion are not fully consistent. Noteworthy, none of the mentioned studies analyzed the rates of malignancy in the subgroup of nodules with nuclear atypia or nodules with architectural atypia separately.
Słowińska-Klencka et al. reported that chances for obtaining category III with repeat FNA are higher in patients with HT than without HT. Unfortunately, the results of the histological examination were not clearly presented. It is not clearly stated, how many patients had thyroid cancer. Only 87 as reported in table S4?
Following that remark we have added a new subsection to the Results which is focused on presentation of histopathological results. This subsection precedes the subsection that describes the analysis of malignancy risk and it includes data on the frequency of surgical treatment in both groups (moved from Material and methods section). We have added a sentence directly reporting numbers of cancers found in both groups. In the original submission these data were given in Table 4 only. Additionally, the paragraph that contained a reference to Table S4 (which describes types of diagnosed cancers in detail) has been moved to the new subsection of the Results.
Furthermore, the results of cytological analysis and the risk of malignancy are not presented clearly. But they are presented several times in different manners in the text and the tables. This makes the text very difficult to understand.
We acknowledge that fact that the amount of information presented about the malignancy risk is larger than usually. That is because we were the first to analyze the malignancy risk in nodules of category III not only in relation to HT status but also to the subcategory of FNA result, i.e. nodules with architectural atypia (FLUS) vs. nodules with nuclear atypia (AUS). Additionally, we assessed the influence of persistent category III outcome in repeat FNA on the risk of malignancy. And the risk of malignancy was evaluated not only in operated patients - what is an estimation of the upper limit of the actual risk - but also in all patients (operated and not operated) – what is an estimation of the lower limit of the risk. Such a comprehensive analysis inevitably brings a lot of numbers to digest but we sincerely hope that they can more easily followed in the revised version.
Reviewer 3 Report
General comments
The study presents very interesting results.
English needs to be revised
The presentation of the results and the discussion could be more summarized.
Comment 1
It is suggested to use the term AUS/FLUS instead of category III of Bethesda system throughout the text
Comment 2
“Category III is the most problematic among these three equivocal categories, as it is characterized by the highest diversity of the estimated risk of malignancy (RoM) which ranges between several and dozens percent [2-9]” (lines 46-49)
It is suggested to quantify the risk according to the indicated references.
Comment 3
“On the other hand, in centers where its RoM is low (<10%) the question arises: should conservative treatment be abandoned when a repeat FNA brings a category III result again (in as much as 1/3 of cases)?” (lines 56-58)
What is the conservative treatment? Surgery? It is suggested to reformulate the sentence.
It is suggested to add reference to the following statements: “in centers where its RoM is low (<10%)” and “in as much as 1/3 of cases”.
Comment 4
“They were shown to have a higher prevalence of equivocal cytology in thyroid nodules when compared to patients without HT [10-12]” (lines 59-61)
It is suggested that the word “equivocal” be replaced by “indeterminate” in the text.
Comment 5
“Our previous analysis indicated that in patients with HT cytological outcomes of category III constituted as much as 15.6% of all FNA and 17.9% of diagnostic FNA, over twice as much as in the general population examined at our center—with or without HT (6.4% of diagnostic FNA) 64 [13]” (lines 61-64)
According to reference 13, the study did not assess HT cytological outcomes but thyroid nodules with Hürthle cells (HC) outcomes. Nodules with HC are not synonymous with HT. According to the same study between 2005 and 2017 and considering only diagnostic cytology, category III was present in respectively 8.6% and 5.2% (P<0.001) of HC smears and non-HC smears. A reformulation of this sentence is suggested.
Comment 6
“The relatively frequent occurrence of the suspicious cytology is a consequence of higher incidence of papillary carcinoma (PTC) in patients with HT” (lines 65-66)
According to a recent review and meta-analysis [Abbasgholizadeh, P., Naseri, A., Nasiri, E. et al. Is Hashimoto thyroiditis associated with increasing risk of thyroid malignancies? A systematic review and meta-analysis. Thyroid Res 14, 26 ( 2021). https://doi.org/10.1186/s13044-021-00117-x] although a significant association was found between HT and some types of thyroid cancers, high risk of bias studies, high level of heterogeneity, and the limited number of well-designed prospective studies, suggested the need for more studies to reach more reliable evidence.
A reformulation of this sentence is suggested.
Comment 7
“We assumed that all patients in HT group had to have a clinical diagnosis of HT confirmed with elevated levels of serum anti-thyroid peroxidase antibodies or characteristic features of HT in the US or microscopic examination (cytological or histopathological)” (lines 85-88)
The definition of HT is very broad and may lead to classification biases. How were the “characteristic features of HT in the US or microscopic examination” defined?
Comment 8
“Exclusion criteria were: (a) positive history of Graves’ disease or increased concentration of antibodies against the TSH receptor, (b) surgical or radioiodine thyroid treatment in the past, (c) positive neck irradiation history” (lines 89-91)
Was the concentration of antibodies against the TSH receptor measured in all participants?
Comment 9
“The surgical treatment was performed based on the patient’s preference or due to the large size of the goiter or the presence of other clinical risk features, including the suspicious ultrasound image of a nodule or as a consequence of the alarming outcome of rFNA” (lines 110-113)
This sentence suggests that the patient may opt for surgery even without clinical indication or that surgery may be indicated only based on ultrasound features. Clarification of this sentence is proposed.
Comment 10
“The mean age of patients in HT group was higher than in non-HT group, as was the percentage of females” (lines 119-120)
Given these characteristics, can age and sex differences interfere with the results? Could these differences be a limitation of the study?
Comment 11
“The percentage of patients with category III diagnosis in more than one nodule was similar in both groups and did not exceed 5% (HT group: 24/4.3% patients with two category III nodules and 3/0.5% patients with three such nodules; non-HT group: 51/4.1% and 4/0.3%, respectively)” (lines 120-123)
What was the percentage of patients with category III diagnosis in only one nodule?
The results referred to, between lines 118 and 131, are presented in an unclear way. Table 1 also does not help to clarify the doubts. Reformulation is suggested.
Comment 12
The significant differences (p<0.001) in relation to the repetition of FNA in the HT and non-HT groups (62% versus 46%) could be better explained.
Why the FNA was not repeated on all Bethesda Category III participants?
Comment 13
"Some patients were operated after rFNA, more often in HT than non-HT group: 35.6% vs. 24.1% (p=0.0060)” (lines 142-143)
The reasons for these differences could be better explained.
Author Response
Thank you for positive comments on our study. The whole manuscript has been rechecked for language errors. We tried to improve the presentation of results following two somewhat contradictory suggestions: to provide some detailed explanations and to make it more summarized. We have addressed all the detailed remarks:
- It is suggested to use the term AUS/FLUS instead of category III of Bethesda system throughout the text
Indeed, in published studies the term “AUS/FLUS” is often used to describe nodules of category III in the Bethesda classification. In our study we evaluated nodules diagnosed with architectural atypia (denoted as FLUS) and those with nuclear atypia (denoted as AUS) separately, as did some other investigators before. Moreover, the distinction proved to be justified as we found significant differences in the risk of malignancy between those two types of nodules. Thus, we deliberately used the term “category III” instead of “AUS/FLUS” as it increased readability of many sentences in which both AUS and FLUS subcategories are being characterized. The interchangeable use of “category III” and “AUS/FLUS” would have led to further misunderstanding. That is why we opted to keep the “category III” term.
- “Category III is the most problematic among these three equivocal categories, as it is characterized by the highest diversity of the estimated risk of malignancy (RoM) which ranges between several and dozens percent [2-9]” (lines 46-49)
It is suggested to quantify the risk according to the indicated references.
Following the suggestion we have shown the exact numbers from the indicated studies, i.e. 2%-81%.
- “On the other hand, in centers where its RoM is low (<10%) the question arises: should conservative treatment be abandoned when a repeat FNA brings a category III result again (in as much as 1/3 of cases)?” (lines 56-58)
What is the conservative treatment? Surgery? It is suggested to reformulate the sentence.
It is suggested to add reference to the following statements: “in centers where its RoM is low (<10%)” and “in as much as 1/3 of cases”.
According to the medical dictionary section of TheFreeDictionary (https://medical-dictionary.thefreedictionary.com) the term “conservative treatment” means a “treatment designed to avoid radical medical therapeutic measures or operative procedures”. So, actually it means an opposite of surgical treatment. However, the sentence has been reformulated to be more clear. The appropriate references have been added to the indicated statements.
- “They were shown to have a higher prevalence of equivocal cytology in thyroid nodules when compared to patients without HT [10-12]” (lines 59-61)
It is suggested that the word “equivocal” be replaced by “indeterminate” in the text.
The suggested replacement has been made to the indicated sentence as well as some other sentences in which categories III and IV of the Bethesda classification are addressed. However, the term “indeterminate” is generally not referred to category V nodules and that is why the term “equivocal” has been preserved in some sentences where that category is being considered.
- “Our previous analysis indicated that in patients with HT cytological outcomes of category III constituted as much as 15.6% of all FNA and 17.9% of diagnostic FNA, over twice as much as in the general population examined at our center—with or without HT (6.4% of diagnostic FNA) 64 [13]” (lines 61-64)
According to reference 13, the study did not assess HT cytological outcomes but thyroid nodules with Hürthle cells (HC) outcomes. Nodules with HC are not synonymous with HT. According to the same study between 2005 and 2017 and considering only diagnostic cytology, category III was present in respectively 8.6% and 5.2% (P<0.001) of HC smears and non-HC smears. A reformulation of this sentence is suggested.
The indicated sentence was imprecise as it actually should include references to our two previous studies not a single one. Data on the frequency of category III in patients with HT come from the paper “The Presence of Hypoechoic Micronodules in Patients with Hashimoto's Thyroiditis Increases the Risk of an Alarming Cytological Outcome. J Clin Med. 2021, 10, 638. doi: 10.3390/jcm10040638.” (reference 22 in the original submission) and that reference has been added (it is reference 26 in the revision). The cited values of 15.6% and 17.9% come from Table 3 of that paper which shows the distribution of the categories of FNA outcomes of the thyroid nodules according to the Bethesda system in 557 patients with Hashimoto’s thyroiditis in relation to the ultrasound pattern of the thyroid parenchyma: 15.6% FNA (87 out of 557) were classified into category III, and they constituted 17.9% FNA (87 out of 485) in which diagnostic specimens were obtained (categories II-VI).
And the data on the frequency of category III in the general population come from the study “Thyroid nodules with Hürthle cells: The malignancy risk in relation to the FNA outcome category. J. Endocrinol. Investig. 2019, 42, 1319–1327” (reference 13 in the original submission). As we stated in that paper “The new category III amounted to 6.4% of all FNA results and was more common among HC nodules (14.4%) than non-HC ones (5.8%)”. The sentence cited by the reviewer refers to years 2005-2017, so a timespan starting before the Bethesda classification was introduced. And we refer in the manuscript to the value of 6.4% that had been calculated for the period of 2010-2017, i.e. the period following the introduction of the Bethesda classification in our country.
- “The relatively frequent occurrence of the suspicious cytology is a consequence of higher incidence of papillary carcinoma (PTC) in patients with HT” (lines 65-66)
According to a recent review and meta-analysis [Abbasgholizadeh, P., Naseri, A., Nasiri, E. et al. Is Hashimoto thyroiditis associated with increasing risk of thyroid malignancies? A systematic review and meta-analysis. Thyroid Res 14, 26 ( 2021). https://doi.org/10.1186/s13044-021-00117-x] although a significant association was found between HT and some types of thyroid cancers, high risk of bias studies, high level of heterogeneity, and the limited number of well-designed prospective studies, suggested the need for more studies to reach more reliable evidence.
A reformulation of this sentence is suggested.
The sentence has been modified to be less definite about the relation between HT and the increased incidence of PTC as follows: “The suspicious cytology may be more common in HT patients because of postulated higher incidence of papillary carcinoma (PTC) in that cohort. Another reason might be a difficulty in the interpretation of microscopic images with coexisting HT due to the com-mon anisocytosis of thyroid follicular cells, their oxyphilic metaplasia, and variable inflammatory infiltration [10-12, 14-15]”.
- “We assumed that all patients in HT group had to have a clinical diagnosis of HT confirmed with elevated levels of serum anti-thyroid peroxidase antibodies or characteristic features of HT in the US or microscopic examination (cytological or histopathological)” (lines 85-88)
The definition of HT is very broad and may lead to classification biases. How were the “characteristic features of HT in the US or microscopic examination” defined?
The indicated statement has been made more specific by adding a reference to another our study (reference 22 in the original submission), where there is a detailed description of US patterns identified in patients with HT at our center. These patterns include: (a) hypoechoic (compared to submandibular glands), homogeneous/fine echotexture; (b) hypoechoic, heterogeneous/coarse echotexture; (c) marked hypoechoic (darker than strap muscles), heterogeneous/coarse echotexture; (d) heterogeneous echotexture with hyperechoic, fibrous septa; (e) multiple, discrete marked hypoechoic areas (micronodules—sized as 1 to 6 mm); (f) normoechoic pseudo-nodular areas; (g) echostructure similar to the connective tissue. We also routinely assess the presence of hypo- or hypervascularity of the gland in Power Doppler imaging, as well as lymph nodes typical of HT. On the basis of our experience and published data, we assume that such nodes show at least one of the following features: round shape, hypoechoic image or absence of the hilum but without features highly suggestive of malignancy: microcalcifications, cystic aspect, peripheral vascularity or diffusely increased vascularization or hyperechoic tissue looking as thyroid and short axis ≥8 mm. Lymph nodes are evaluated at infrathyroidal and pretracheal areas and along the carotid arteries and jugular veins.
We consider the following cytological image as typical of HT: inflammatory cells ( lymphocytes, plasmacytes predominating in the smears) and thyroid follicular cells (usually scattered or clustered in small groups) showing anisocytosis and frequently oxyphilic metaplasia. In majority of such cases the cytological report was concluded with the statement indicating the presence of microscopic features of HT in a smear. An adequate statement has been added to the Material and methods section.
- “Exclusion criteria were: (a) positive history of Graves’ disease or increased concentration of antibodies against the TSH receptor, (b) surgical or radioiodine thyroid treatment in the past, (c) positive neck irradiation history” (lines 89-91)
Was the concentration of antibodies against the TSH receptor measured in all participants?
No, anti-TSH receptor antibodies were not measured routinely, but if such a test had been performed its results were used.
- “The surgical treatment was performed based on the patient’s preference or due to the large size of the goiter or the presence of other clinical risk features, including the suspicious ultrasound image of a nodule or as a consequence of the alarming outcome of rFNA” (lines 110-113)
This sentence suggests that the patient may opt for surgery even without clinical indication or that surgery may be indicated only based on ultrasound features. Clarification of this sentence is proposed.
The FNA diagnosis of category III is not regarded as a routine indication for the surgical treatment, at least in our center. However, it poses an increased risk of thyroid malignancy and the patients are informed about it. Some patients, with an increased level of carcinophobia, find it difficult to cope with such a diagnostic uncertainty and opted for surgical treatment. Accordingly, the surgical treatment may be justified in such cases. We believe that the sentence is clear enough, but it must be interpreted in the context of the studied population, which consisted of patients with category III diagnosis only.
- “The mean age of patients in HT group was higher than in non-HT group, as was the percentage of females” (lines 119-120)
Given these characteristics, can age and sex differences interfere with the results? Could these differences be a limitation of the study?
The difference in the mean age of patients between HT and non-HT groups was 2 years and several months. It was significant (due to the large number of examined patients) but relatively small. Thus, we are pretty sure that it is not a limitation of our study. Notably, we did not observed any significant differences in patient’s age between surgically treated patients with cancers and benign lesions, either in the whole examined cohort or in HT and non-HT groups taken separately.
Similarly, the percentage of women was significantly higher in HT group than in non-HT group only when the whole cohort was evaluated. Again, there was no significant difference between surgically treated patients with cancers and benign lesions, either in the whole examined cohort or in HT and non-HT groups taken separately.
Adequate statements have been added to the Results section.
11., “The percentage of patients with category III diagnosis in more than one nodule was similar in both groups and did not exceed 5% (HT group: 24/4.3% patients with two category III nodules and 3/0.5% patients with three such nodules; non-HT group: 51/4.1% and 4/0.3%, respectively)” (lines 120-123)
What was the percentage of patients with category III diagnosis in only one nodule?
The results referred to, between lines 118 and 131, are presented in an unclear way. Table 1 also does not help to clarify the doubts. Reformulation is suggested.
Table 1 clearly indicates that the percentage of patients with more than one nodule of category III was 4.8% in HT group and 4.4% in non-HT group. According to the study design all patients had nodules with the diagnosis of category III. So it is possible to calculate that in HT group the percentage of patients with category III diagnosis in only one nodule was 100% - 4.8% = 95.2% in HT group and 100% - 4.4% = 95.6% in non-HT group. We believe that such data in the text would be a redundancy, and it would not be justified by their relevance. We mentioned patients with more than one category III nodule only to indicate the reason why the numbers for per-patient and per-nodule analyzes are slightly different.
- The significant differences (p<0.001) in relation to the repetition of FNA in the HT and non-HT groups (62% versus 46%) could be better explained.
Why the FNA was not repeated on all Bethesda Category III participants?
We addressed that in the Discussion with the following sentence: „So our observation of a higher frequency of the surgical treatment performed directly after first diagnosis of category III in patients without HT was probably a consequence of larger nodules’ sizes and not an especially suspicious ultrasound image. – lines 397-399 of the original submission.
Our study was a retrospective one. We did not assume to plan particular diagnostic or therapeutic steps for examined patients. We can only speculate on actual causes for undertaking surgical treatment just after the first diagnosis of category III. We believe, accordingly to what we wrote, that the important reason could be a large goiter or large nodule’s size (data on nodules’ sizes in both groups were shown in Table1). As we indicated in another part of the discussion our population had been exposed to iodine deficiency for many years (lines 339-340 of the original submission) and multinodular goiter still remains a frequent indication for surgical treatment. Obviously, repeat FNA was not performed if a patient did not give their consent or was lost from observation (as indicated in Figure S1).
- "Some patients were operated after rFNA, more often in HT than non-HT group: 35.6% vs. 24.1% (p=0.0060)” (lines 142-143)
The reasons for these differences could be better explained.
The explanation is at least partially analogous to our comment to remark 12. In HT group surgical treatment was performed directly after first diagnosis of category III less often than in non-HT group because of smaller nodules. Accordingly, among patients treated surgically the frequency of patients operated after repeat FNA was higher in HT group than in non-HT group (35.6% vs 24.1%).
Interestingly, there was no significant difference in rates of surgical treatment in patients after repeat FNA between analyzed groups. The percentage of operated patients among those who had repeat FNA performed was actually the same in both groups and equaled to 16.3% (57 out of 349 in HT group and 94 out of 575 in non-HT group). That was the case despite repeat FNA diagnosed category III more often in HT group than non-HT group. But at the same time in non-HT group repeat FNA more often brought results category I or IV.
The indicated sentence has been rewritten in the revised version and moved to the new section 3.2. “Surgical treatment without or after repeat FNA”. We believe that extending discussion on the reasons behind mentioned differences is not necessary as they are not really relevant to the aim of study and its conclusions. Again, that differences reflect intrinsic limitations of a retrospective study.
Round 2
Reviewer 2 Report
Słowińska-Klencka et al. aimed to evaluate the risk of malignancy (RoM) in category III thyroid nodules of Bethesda system in patients with and without Hashimoto thyroiditis (HT) and to analyze whether obtaining category III with a repeat FNA (rFNA) increases RoM. The study included 563 HT and 1250 non-HT patients. A repeat FNA was performed in 349 and 575 patients, while surgical treatment was done in 160 and 390 patients, respectively. As many other authors also Słowińska-Klencka et al. found out, that the frequency of malignancy was not different in patients with and without HT. Słowińska-Klencka et al. reported that chances for obtaining category III with repeat FNA are higher in patients with HT than without HT. Unfortunately, also in the reviewed manuscript the same shortcomings remain as were there in the first version of the manuscript. The text is not written transparently and is therefore not understandable to the reader. The results are repeated in tables and in several places in the text, which makes it impossible for the reader to understand the text. The authors try to present and analyze several different observations, which makes the article completely opaque and therefore not understandably written. Furthermore, there remain a big possibility of selection bias for repeat FNA biopsy and even bigger for surgical procedure and risk of malignancy. My view is that the manuscript should not be published in the Cancers journal.
Reviewer 3 Report
No comments